# Immunoinformatics and Reverse Vaccinology Approach for the Identification of Potential Vaccine Candidates against *Vandammella animalimors*

**DOI:** 10.3390/microorganisms12071270

**Published:** 2024-06-22

**Authors:** Ahmad Hasan, Wadi B. Alonazi, Muhammad Ibrahim, Li Bin

**Affiliations:** 1State Key Laboratory of Rice Biology and Breeding, Key Laboratory of Molecular Biology of Crop Pathogens and Insects, Institute of Biotechnology, Zhejiang University, Hangzhou 310058, China; ahassancui@gmail.com (A.H.); ibrahim.zju@gmail.com (M.I.); 2Health Administration Department, College of Business Administration, King Saud University, Riyadh 11421, Saudi Arabia; waalonazi@ksu.edu.sa

**Keywords:** *V. animalimorsus*, reverse vaccinology, subtractive genomics, drug targets, MHC-I, MHC-II T-cell, B-cell epitopes, chimeric vaccine, pET-28a (+) vector

## Abstract

*Vandammella animalimorsus* is a Gram-negative and non-motile bacterium typically transmitted to humans through direct contact with the saliva of infected animals, primarily through biting, scratches, or licks on fractured skin. The absence of a confirmed post-exposure treatment of *V. animalimorsus* bacterium highlights the imperative for developing an effective vaccine. We intended to determine potential vaccine candidates and paradigm a chimeric vaccine against *V. animalimorsus* by accessible public data analysis of the strain by utilizing reverse vaccinology. By subtractive genomics, five outer membranes were prioritized as potential vaccine candidates out of 2590 proteins. Based on the instability index and transmembrane helices, a multidrug transporter protein with locus ID A0A2A2AHJ4 was designated as a potential candidate for vaccine construct. Sixteen immunodominant epitopes were retrieved by utilizing the Immune Epitope Database. The epitope encodes the strong binding affinity, nonallergenic properties, non-toxicity, high antigenicity scores, and high solubility revealing the more appropriate vaccine construct. By utilizing appropriate linkers and adjuvants alongside a suitable adjuvant molecule, the epitopes were integrated into a chimeric vaccine to enhance immunogenicity, successfully eliciting both adaptive and innate immune responses. Moreover, the promising physicochemical features, the binding confirmation of the vaccine to the major innate immune receptor TLR-4, and molecular dynamics simulations of the designed vaccine have revealed the promising potential of the selected candidate. The integration of computational methods and omics data has demonstrated significant advantages in discovering novel vaccine targets and mitigating vaccine failure rates during clinical trials in recent years.

## 1. Introduction

Rabies is currently known as the leading viral zoonosis due to its widespread global occurrence, elevated incidence rates, significant veterinary and human tolls, and its high mortality rate, leading to substantial economic burdens in many countries [1,2]. Resulting from bites, rabies demonstrates not only a virulence increase but also a notable fatality rate. It also presents a severe viral illness inducing a neurological disorder in humans, mammals, and also in warm-blooded animals [3,4]. Approximately 4.5 million individuals globally are masticated by animals every year according to the Centers for Disease Control and Prevention and usually require post-exposure prophylaxis [5]. As per data from the World Health Organization, epidemiological evidence unveils that probably 2.5 billion people have been exposed to the risk of rabies, and around 50K-60K deaths are reported annually associated with rabies, of which 31–32,000 occur in Asia and Africa. Each year, around 10 million people undertake post-exposure rabies vaccination [6,7,8].

Human animal bite wound infections are generally triggered by a polymicrobial microbiome that includes both anaerobic and aerobic microbes. The oral flora of the animal that bit the victim is typically reflected in the bacteria identified in these infected wounds. The diet of the animal and other items it consumes can have an impact on this flora, and also, at the moment of the injury, bacteria could have originated from the victim’s skin or the surrounding area [9,10]. Human injuries resulting from animal bites can lead to punctures, cuts, and subsequent bacterial and fungal infections. Each bite introduces microorganisms into wounds, posing a significant risk to the integrity of human skin, muscles, and even bones [9]. 

The infectious risk due to animal bites primarily arises from saprophytic pathogens in the animal’s oral cavity and on its skin. The most commonly isolated strains include *Streptococcus pyogenes*, *Staphylococcus intermedius*, *Streptococcus viridans*, *Neisseria* spp., *Fusobacterium* spp., *Moraxella* spp., *Bacteroides fragilis*, *Prevotella* spp., *Porphyromonas* spp., *Capnocytophaga canimorsus*, and *Pasteurella multocida* [10,11]. The infection with dog bite, onset within 24–48 h, manifests with edema and pain at the inoculation site, followed by maculopapular rash, cellulitis, abscesses, and complications such as osteomyelitis, meningitis, pneumonia, and renal failure with sepsis. Risk factors that expressively increase mortality, up to 25%, include asplenia and immunosuppression [11]. 

*V. animalimorsus* is a Gram-negative bacterium that belongs to the family *Comamonadaceae* and is provisionally named *V. animalimorsus* when recovered from human wound infections, primarily after animal bites. The bacterium has gained attention as an emerging pathogen, particularly in cases of wounds caused by dog or cat bites [12]. This bacterium can cause serious infections, especially in immunocompromised individuals, leading to complications such as septicemia, endocarditis, and other severe conditions [13,14]. 

The reverse vaccinology technique, a component of the vaccinomics regime, employs computational biology to examine entire pathogen genomes to identify proteins that could yield promising epitopes, peptides within an antigen where antibodies bind, and surface-situated proteins. This method has been extensively utilized to prioritize and develop vaccine targets against various infectious pathogens, such as *E. coli*, *Mycobacteroides abscessus*, *Acinetobacter baumannii*, Yellow fever etc., as reviewed by Jalal et al. [15]. The lack of potential antibiotic resistance and vaccines [12,13] highlights the need for continued research and preventive measures to manage and mitigate the risks associated with this bacterium. The development of a vaccine effective against *V. animalimorsus* could provide significant assistance for public health and the economy, given the rising antibiotic resistance among *V. animalimorsus* strains and the resulting impacts on mortality, morbidity, and productivity loss.

This study was designed to analyze the complete proteome of *V. animalimorsus strain* NML00-0135 and identify potential vaccine candidates using subtractive genomics and reverse vaccinology. Subsequently, we mapped and compiled immunogenic epitopes from these candidates to construct a multi-epitope vaccine. This vaccine then underwent a computational assessment to evaluate its physicochemical, chemical, and immunological properties, with the aim of selecting it as a potential vaccine candidate against pathogenic *V. animalimorsus*. 

## 2. Materials and Methods

The comprehensive and extensive in silico method was opted to carry out the prediction and rationality of the potential vaccine candidates. The graphical representation of the whole work is described in Figure 1.

### 2.1. Proteome Data Acquisition and Finding Non-Redundant Proteins

Out of eight entries associated with *V. animalimorsus,* the complete proteome of the *V. animalimorsus* strain NML00-0135 reference strain resourced from human dog bite infection was retrieved from the Universal Protein Resource Knowledgebase (UniProtKB) [16]. The Cluster Database at High Identity with Tolerance (CD-HIT) was utilized to identify the non-paralogue proteins [17]. Proteins demonstrating over 80% similarity with other proteins are believed to be paralogous, setting a threshold of 0.8, and subsequently, they were excluded, leading to the identification of exclusively non-paralogous proteins.

### 2.2. Prediction of Non-Homologues and Essential Proteins

The non-homologous proteins are considered as potential vaccine candidates, and such proteins could be useful to avoid cross-reactivity. The resulting protein sequences from the CD-HIT analysis underwent BLASTp using NCBI against the entire proteome of the *Homo sapiens* with a cutoff E-value of 10^−3^ and other default parameters. The BLASTp analysis yielded “Hits” of homologous sequences with over 80% similarity to humans, and in the case of non-homologous sequences, resulted “No Hits”. Moreover, essential proteins that play a crucial role in cell metabolism and are indispensable for the survival of an organism were further retrieved. The non-homologous sequence was then analyzed to filter the essential proteins by BLASTp analysis of Database of Essential Gene (DEG) [18], setting E-value 10^−5^ as the cut-off value and further default values. The DEG contains experimentally identified essential genes. A significant number of the gene sequences were retrieved from the DEG for further analysis, excluding the non-similar proteins. DEG provides researchers with a curated and annotated collection of essential genes across various organisms, such as bacteria, yeast, and even higher organisms like humans.

### 2.3. Prediction of Cytoplasmic and Outer Membrane Proteins

To design a reliable multi-epitope vaccine construct, the determination of the subcellular localization of proteins such as selection of extracellular or outer membrane proteins is imperative. The subcellular localization of proteins was examined by using PSORTb version 3.0.29 [19] and CELLO bioinformatics tools. These tools predict the subcellular localization of proteins by analyzing their amino acid sequence. Possible subcellular localization results may include unknown localization, cell wall, cytoplasmic membrane, and extracellular proteins. 

### 2.4. Prioritization of Antigenic Proteins

Antigens are molecules that are recognized by the host immune system to elicit the immune response [15], whereas intracellular proteins can be targeted for potential drug targets. The outer membrane contains numerous proteins that have been known as immunogenic and hold potential as vaccine targets [20]. The predicted outer membrane protein was further processed for prioritization of antigenicity. The VaxiJen v2.0 server was utilized to predict the most potent antigenic proteins from the whole *V. animalimorsus* proteome with default parameters of 0.5 [21].

### 2.5. Epitope Mapping and Population Coverage

The Immune Epitope Analysis Database (IEDB) was utilized to map the epitopes of the protein sequences that were considered final and shortlisted, because they meet the criteria for possible candidates in a multi-epitope peptide vaccine against *V. animalimorsus* [17]. The linear epitopes were chosen due to their capability to bind to antibodies even following denaturation. The anticipated B-cell epitope was then utilized in predicting multiple allele MHC-II binding epitopes, and MHC-I binding epitopes were predicted using MHC-II binding epitopes as input sequences. For T-cell epitopes (MHC-I and MHC-II binding), a comprehensive reference set of alleles was chosen. The predicted epitopes were analyzed through MHCPred [22], and the criteria for selection of epitopes were set below 100 nM IC50 values. The epitopes selected were further examined for allergenicity, toxicity, antigenicity, and solubility. The tools used to assess toxicity and solubility were ToxinPhred [23] and Protein-Sol, respectively. Finally, only those epitopes determined to be soluble, nontoxic, antigen-free, and nonallergenic were selected. The IEDB tool was used to evaluate the population coverage of the fourteen shortlisted epitopes [24]. Rather than focusing on a specific region, population coverage studies use MHC-I and II alleles on the entire world population. This is in line with plans to develop a vaccine that will target different groups around the world. MHC class I and class II binding alleles were used in this study.

### 2.6. Multi-Epitope Vaccine Design

The MHC-1 epitopes with potential vaccine candidacy were linked using GPGPG linkers [25], while MHC-II epitopes were linked using the EAAAK linker [26]. It is well known that the addition of an adjuvant increases the overall immunogenicity of a multi-epitope peptide [27]. The GPGPG linkers can enhance efficient immune processing and facilitate epitope presentation to the immune system; they were chosen for epitope peptide construction. It is important to mention that the linkers used in this study have better flexibility and structural stability with reduced immunogenicity and protease resistance compared to other linkers [25,26,27]. The adjuvant was first linked to the CTL epitope using EAAAK linkers, which required isolation of the domains in bipartite fusion proteins [28,29]. In the multi-epitope’s peptide *V. animalimorsus* vaccine, the Cholera Toxin B (CTB) subunit served as the adjuvant as CTB has been studied as a traditional mucosal adjuvant to boost vaccination immunogenicity [30]. The biochemical features of the vaccine construct such as instability index, molecular weight, hydrophilicity, and therapeutic index were obtained using Protparam [31]. Furthermore, I-TASSER [32] was utilized for the prediction of the 3D structure of the vaccine construct designed with epitopes. Following the determining of the 3D model of the vaccine construct, the process continued with refinement and loop modeling using Galaxy Loop and Galaxy Refine within Galaxy Web [33].

### 2.7. Codon Optimization and In Silico Cloning of Final Vaccine Construct

To back-translate the vaccine sequences into cDNA, the Java Codon Adaptation Tool (JCAT) was employed, with the aim of enhancing the expression of the designed vaccine. This tool facilitates the determination of the GC contents as well as the Codon Adaptation Index (CAI) score of the DNA sequence for the optimized nucleotide sequence. Throughout the optimization process, care was taken to prevent Rho-independent termination of transcription cleavage sites for restriction enzymes and prokaryotic ribosome binding sites [34]. Additionally, SnapGene, a molecular cloning and sequence analysis software v 7.2 developed by Insightful Science, was used. SnapGene allows biologists to easily visualize, analyze, edit, and share molecular biology sequences and procedures.

### 2.8. Computational Immune Simulation and Molecular Docking Analysis

The C-ImmSim online server enables users to designate the antigen for injection using a list of multi-protein FASTA text, Uniprot accession numbers and PDB primary identifiers. Haplotypes can be specified through additional parameters, such as volume, simulation time, and through drop-down menus as outlined in the referenced work [35]. By using the structures of the individual proteins involved, computational docking provides an alternative method for predicting the structure of protein–protein complexes. Although X-ray crystallography is generally considered more reliable, the computational approach produces models that can be verified by more straightforward experimental methods such as site-directed mutagenesis or cross-linking. ClusPro, one of the most widely used protein–protein docking servers out there, was employed, developed by the Boston University Wijda Lab [36].

### 2.9. Molecular Dynamics Simulation

A method frequently used in computational research to assess the stability of protein–protein complexes is normal mode analysis (NMA) [37]. This analytical method is utilized to examine the dynamics of proteins, and their movements are contrasted with their specific modes [38]. NMA is a protein mobility approach which inspects the motions of specific proteins within their internal coordinates [39]. In our analysis, we used the iMODS server for this purpose [40,41], which provides insightful information on the fundamental dynamics of many proteins [42]. The tool predicted protein dynamics and stability using several metrics such as eigenvalues, covariance, B-factors, and deformability [43]. Eigenvalues show how conformable the main chain of the protein is, i.e., lower values indicate that the protein structure is more prone to bending or distortion, as it has a direct effect on the amount of energy required for these adjustments. Similarly, coherence analysis shows how different regions of a protein move together to help in detecting coordinated movements within the protein structure [44]. Elevated B factors indicate greater adaptability and mobility. NMA investigation of the dynamics of protein–protein complexes is facilitated by the iMODS server [45]. This knowledge is crucial for predicting the behavior of the complex and its suitability for the intended use, such as the creation of vaccines or other medicinal projects [46].

## 3. Results and Discussion

The proteome of *V. animalimorsus* strain NML00-0135 reference strain obtained from UniProt initially comprised a total of 2590 proteins. The core proteome was acquired and employed as potential targets for vaccine design against the pathogen. To subtract the proteomics data and to remove the redundant proteins, the proteome was subjected to CD-HIT clustering with a 90% threshold, which resulted in the removal of duplicate sequences and the retention of only non-redundant 2574 proteins. Proteins resulting from gene duplication, known as redundant proteins, exist in multiple copies within genomes/proteomes. Therefore, their inclusion in subsequent analyses may be scientifically less sound. On the contrary, non-redundant proteins, fundamental in pathogen essential pathways and functions, emerge as potential vaccine targets [47].

### 3.1. Prediction of Non-Homologues and Essential Proteins

To eliminate the host homologs from the non-redundant proteome, BLASTp was employed against the human proteome, resulting in the identification of 785 proteins as host non-homologous proteins. The presence of host non-homologous proteins reduces the risk of triggering auto-immune reactions and enhances the safety profile of the selected proteins for further analysis [47,48]. Essential proteins are fundamental for maintaining cellular functions and constitute a minimal set of proteins required for life to sustain [43]. BLASTp was run against the DEG, which revealed 698 as essential proteins. Essential proteins play pivotal roles in regulating critical mechanisms such as virulence, nutrient uptake, and pathogenicity [49,50,51] and are potential candidates for the vaccine development.

### 3.2. Sub-Cellular Localization Assessment

Essential proteins lacking homology were further analyzed based on their localization, recognized as a critical factor for assessing the vaccine suitability. During this phase, only essential proteins retrieved as outer membrane were chosen for further investigation. A total of five proteins were identified as outer membrane using CELLO server and PSORTb (Table 1). Anticipating the cellular location of unidentified proteins aids in knowing their functions, connection to disease processes, and the creation of innovative medications. Gram-negative bacteria possess outer membrane (OM) proteins recognized as promising vaccine candidates. Antigenic diversity among strains has hindered vaccine development, but conserved OMPs across serotypes show potential, and various studies have demonstrated their efficacy and promise [47]. The identification of the five essential OM proteins seems to have potential vaccine candidacy and were processed further. 

### 3.3. OM Proteins Analysis for Chimeric Epitope-Based Vaccine Design

The OM proteins following the analysis of the presence of transmembrane helices were considered acceptable, having transmembrane helices less than or equal to 1. The proteins having transmembrane helices more than 2 were excluded and not indicated as good candidates for vaccine design [52]. 

All five core proteins named > A0A2A2AF83 Glutamate dehydrogenase OS = *V. animalimorsus*, >A0A2A2AIP1 Outer membrane protein assembly factor BamA OS = *V. animalimorsus*, >A0A2A2AJA1 TonB-dependent siderophore receptor OS = Vandammella, >A0A2A2APV4 Type IV pilus biogenesis/stability protein PilW OS = *V. animalimorsus* (Table 1), and >A0A2A2AHJ4 Multidrug transporter OS = *V. animalimorsus* have transmembrane helices less than 1 or zero as described in Table 1 and were selected. Among them, one protein >A0A2A2AHJ4 Multidrug transporter OS = *V. animalimorsus* was selected, having the best instability indexes [53] of less than 50 and more than the other four proteins’ 49.20. Moreover, the protein was nonpolar based on hydrophilicity and having a negative GRAVY value and was also predicted as nonallergenic. Antigenicity was also checked and found to be antigenic with antigenicity values of 0.5390. This core protein met the criteria for potential vaccine candidates while also having no significant similarity to human or Lactobacillus species bacteria using multidrug transporter proteins, a type of efflux pump, to expel a wide range of antibiotics and other toxic substances from their cells [54]. By inducing an immune response against these proteins, it may be possible to enhance the efficiency of prevailing antibiotics and reduce the prevalence of multi-drug-resistant infections.

### 3.4. Epitope Prediction and Prioritization Leading to Vaccine Design

The A0A2A2AHJ4 Multidrug transporter protein listed as a potential vaccine candidate was employed to determine B-cell, MHC-I, and MHC-II epitopes for constructing a chimeric vaccine targeting *V. animalimorsus*. Several combinations of adaptable adjuvant peptides and immune boosters were applied to link primary epitopes through appropriate epitope-specific linkers. B-cell epitope prediction was conducted using the B-cell epitope prediction by IEDB, resulting in sixteen B-cell epitopes.

Moreover, resultant B-cells epitopes were further examined and shortlisted on the basis of BepiPred linear epitope prediction (Figure 2A), Chou–Fasman beta-turn prediction (Figure 2B), Kolaskar Tongaonkar antigenicity (Figure 2C), Emini surface accessibility prediction (Figure 2D), Karplus–Schulz flexibility prediction (Figure 2E), and Parker hydrophilicity prediction (Figure 2F). The bindings of B-cell epitopes to antibodies lead to the activation of the humoral immune responses [55]. Further screening was carried out on the predicted peptides identified as B-cell epitopes to determine their potential as T-cell epitopes and to determine the presence of binding sites for both MHC class I and class II molecules [56]. 

Sixteen B-Cell epitopes were submitted to predict MHC-II epitopes. These epitopes were analyzed by MHC pred to sort out epitopes inclining to bind with the allele DRB 0401, as this allele is prevalent across the human population, being widely distributed. Epitopes exhibiting strong binding interactions with the DRB 0401 allele provoke strong immunological responses [57]. The epitopes with IC50 values below 100 yielded 135 MHC-I and 46 MHC-II molecules suitable for designing vaccine constructs. Furthermore, 55 epitopes of MHC-I and 18 MHC-II were found as allergens when scrutinized for allergenicity by AllerTop v.2.0. 

Antigenicity evaluation was carried out on the remaining 80 MHC-I and 28 MHC-II nonallergen epitopes by utilizing Vaxijen (at threshold 0.5) to assure that the epitopes are adequately strong to bring immunogenic responses. This resulted in 35 MHC-I epitopes and 17 MHCI-II antigenic epitopes followed by a toxicity check and removing the toxic epitopes, which resulted in 27 MHC-I epitopes and 14 MHC-II non-toxin epitopes proceeded to the solubility test, which found that 8 MHC-I epitopes, 7 MHC-II, and 16 B-cell epitopes were good soluble epitopes and considerable for vaccine design. The details of these features are described in Table 2 and aligned with previous studies [58,59]. 

Vaccines designed with antigenic proteins specific to these pathogens possess the ability to provoke specific immune responses, targeting conserved epitopes within the entire antigen while avoiding reactions to non-neutralizing epitopes that may trigger immunopathogenic or immune-modulating consequences [59]. Thirty-one epitopes meeting the criteria of potential immunological activity and safety were linked using GPGPG linkers. These linkers, being rigid, facilitate the effective separation of epitopes. Cholera toxin B subunit which served as an adjuvant was linked with epitopes by the EAAAK linker. The EAAAK linker serves to keep the linked epitopes and adjuvant separate, preventing their folding over each other (Figure 3). Typically, adjuvants are employed to improve the immune stimulation of vaccines, producing improved immune responses such as the induction of antigen presentation, chemokines, and cytokines [60].

The multi-epitope vaccine construct was examined for allergenicity and antigenicity and revealed to be antigenic with an antigenicity of 0.6718 and with a molecular weight of 67,469.20, and a GRAVY equal to −0.410 is an indication of being hydrophilic. The instability index (II) is computed to be 49.06, declaring the range at which the vaccine is believed as safe within the multi-epitope sequence; individual epitopes are linked together using GPGPG linkers (shown in cream), and AAY linkers (depicted in brown) connect these epitopes (Figure 3). This design ensures a structured and organized arrangement, facilitating the potential for a comprehensive immune response by stimulating both B-cell and T-cell immunity [61].

### 3.5. Structure Modelling and Docking Analysis of Vaccine Construct

The potential vaccine construct sequence was submitted to I-TASSER for 3D structure prediction, yielding seven hypothetical 3D structure models. The criteria for choosing the best model involved a high percentage of residues located within the most favored region of the Ramachandran plot as well as the lower z-score. A lower z-score proposes that the 3D refined model of the vaccine is of higher quality [62]. In molecular docking, TLR4 serves as the receptor for interaction with the formulated vaccine and plays a crucial role in immune activation and the initiation of cytokine responses [63]. Models resulting from the docking process were organized and ranked according to clusters, considering various coefficients indicating distinct energies. The initial model was chosen due to its possession of the lowest binding energy.

A molecular docking simulation was executed on the ClusPro web server, resulting in the identification of the lowest energy state as well as the sbk8. The sbk8 structure served as a receptor in the molecular docking process of the designed vaccine (Figure 4A). The results of H-bond analysis indicated that specific residue pairs interact with each other, including SER 473- LYS 109, GLU 849- THR 112, GLU 850- THR 112, HIS 1031- GLU 111, ASP 1445- ARG 106, ARG 1650- ASP 100, LYS 1876- ASP 101, ASP 1891- THR 115, ASP 1892- SER 103, and ARG 2483- ASP 101, with distance 3.09, 3.19, 3.01, 3.29, 2.65, 2.82, 3.27, 2.64, and 2.70, as shown in Figure 4B. Moreover, sbk8, a Protein Data Bank (PDB) entry, represents a specific protein structure utilized in the docking analysis (Figure 4B). PDBsum, a pictorial database, provided an overview of the 3D structure deposited in the PDB. We used the PDB-sum for illustration of different interactions. This shows that the vaccine is very effective in triggering an immune response because it binds strongly to the TLR4 receptor on immune cells as shown in Table 3.

It displayed the components of the structure, such as metal ions, ligands, protein chains, and DNA along with schematic diagrams illustrating their interactions. A Ramachandran plot analysis was conducted on a protein structure comprising 1356 residues, excluding glycine and proline residues, which totaled 1231 after refinement. 

The analysis revealed that 74.7% of the residues were located in the most favored regions [A, B, L], while 22.0% were in additional allowed regions [a, b, l, p]. Only a small portion, 2.0%, fell into generously allowed regions [~a, ~b, ~l, ~p], with 1.2% of residues being in disallowed regions [XX]. Additionally, there were nine end residues excluding glycine and proline, along with 66 glycine residues and 50 proline residues. This comprehensive analysis provides insights into the conformational preferences and structural stability of the protein under investigation. 

The G-factor analysis provides insights into various structural parameters of the protein [59]. For dihedral angles, the distribution of phi-psi angles showed a score of −0.75, indicating some deviation from the ideal distribution. The chi1-chi2 distribution scored −0.20, while chi1 alone was nearly ideal at −0.01. However, the chi3 and chi4 angles scored positively at 0.58, suggesting a favorable distribution. The omega angle scored −0.22, indicating a slight deviation.

In terms of main-chain covalent forces, the main-chain bond lengths scored positively at 0.16, suggesting adequacy in this aspect. However, main-chain bond angles scored −0.52, signifying some deviation (Figure 5 and Table 4). The overall average G-Factor for these parameters was −0.24, indicating some structural deviations overall. This comprehensive analysis aids in understanding the protein’s structural stability and potential areas for improvement.

### 3.6. Population Coverage

The population coverage for the selected four epitopes was evaluated using the IEDB tool, considering MHC-I and II alleles across the global population. This in-depth analysis encompasses worldwide coverage, aligning with the strategy of designing a vaccine to target diverse populations globally. The binding allele set of MHC-I and II used for the analysis is included. The world population coverage analyzed by IEDB for MHC- I and II was 70.8% and 26.12%, respectively, against the epitopes considered as potential candidates for the vaccine to be designed, as depicted graphically in Figure 6A,B, while the global combined population is 78.4%, represented in Figure 6C. Population coverage, in the context of epitope prediction and vaccine design, indeed refers to the likelihood or probability of epitopes binding to MHC molecules across diverse populations worldwide [64].

### 3.7. Codon Optimization and In Silico Cloning

The codon optimization was done by employing the JCAT tool and cloning of the final vaccine. The predicted average Codon Adaptation Index (CAI) value and GC content for the final vaccine were 56.09% and 0.94, respectively (Figure 7). These results suggest that the vaccine construct has been effectively expressed within the *E. coli* system, indicating successful implementation of the expression process [65]. Subsequently, the SnapGene tool was utilized to incorporate the adapted codon sequence of the final vaccine into the pET28a (+) vector for the construction of the recombinant plasmid (Figure 8).

### 3.8. Computational Immune Simulation

A computational simulation of the immune system was conducted to assess virtu-al immune responses to the formulated vaccine. The findings demonstrated its effectiveness in eliciting immune responses, evidenced by the generation of immunoglobulins. Additionally, the simulation revealed the induction of IFN-g (a cytokine) and interleukins. Figure 9A demonstrates the production of cytokines, which are computational immune responses to the vaccine. Over time, antibody production escalated from primary to secondary stages, underscoring its crucial role in clearing pathogens, as depicted in Figure 9B. Notably, Figure 9C highlights a substantial increase in the production of Plasma B cells, underscoring its crucial role in clearing pathogens.

### 3.9. MD Simulation and Analysis

The docked complex displayed minimal deformation and enhanced rigidity during the dynamic simulation; it reveals that the predicted binding orientation of the ligand within the receptor remained stable and that the complex maintained its structural integrity over the course of the simulation [65,66]. NMA performed by iMODS on the MEV-receptor docked complex provided insight into the disorder of the residues. The peaks in the disorder graph indicate how easily the structure of the protein can change. Comparing the B-factor graphs between the PDB and NMA data shows higher peaks in NMA, suggesting a higher elasticity predicted by computational simulations than observed experimentally. The energy required to deform a structure is quantified by its eigenvalue. Lower values on the graph indicate that only a minimal amount of energy is required to correct the structure, thus confirming significant flexibility and stability of molecular motion. The variance graph shows a correlation with the eigenvalue graph. Red represents individual variation, while green represents cumulative variation. The covariance matrix illustrates the relationship between residuals. Red, white, and blue colors indicate correlated, unrelated, and conflicting experiences of amino acids, respectively. The docking complex exhibits enhanced correlation between pairs of residues, thus ensuring the stability of the docking complex (Figure 10).

## 4. Conclusions

By targeting outer membrane proteins and using reverse vaccinology, a new multi-epitope vaccine candidate is designed with favorable antigenicity, hydro flexibility, and structural stability. The potential vaccine candidate indicates the potential effectiveness of vaccines in globally diverse populations following population coverage analysis, against *V. animalimorsus* infection. Molecular docking simulations have identified potential interactions between formulated vaccines and host receptors, shedding light on the underlying mechanisms of vaccine efficacy and stability. Since the application of bioinformatics tools in the production of epitope-based antigens has become a potential strategy to acquire a novel and powerful vaccine against infectious agents, the construction of single- or multi-epitope-based antigens expressing potential B- or/and T- cell epitopes would greatly improve *V. animalimorsus* immunization strategies. Further experimental validation and clinical studies are guaranteed to validate the efficacy and safety of designed interventions, ultimately leading to clinical implementation.

## Figures and Tables

**Figure 1 microorganisms-12-01270-f001:**
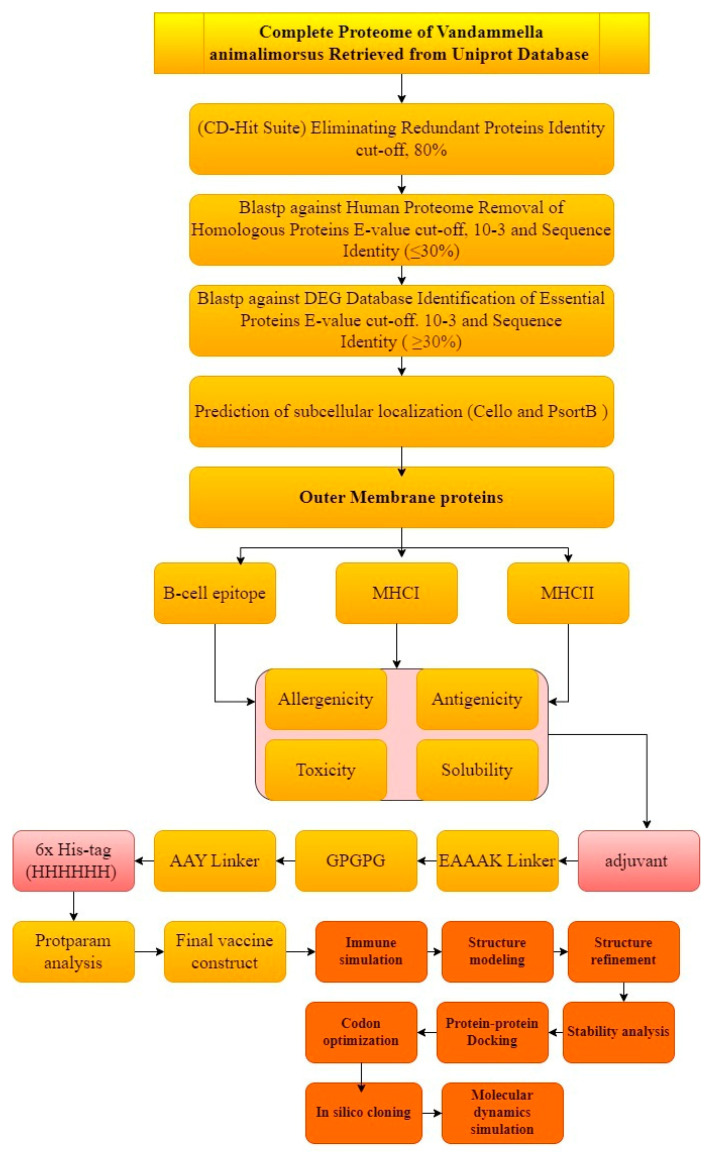
Schematic representation of methodology.

**Figure 2 microorganisms-12-01270-f002:**
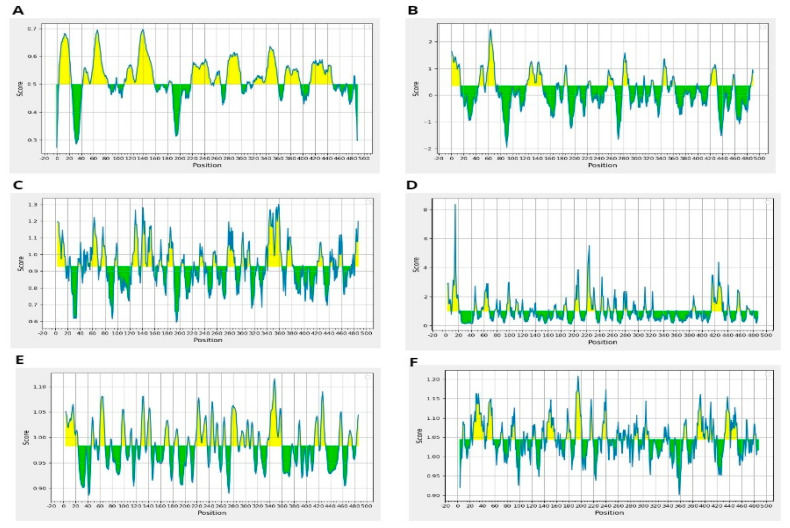
(**A**) BepiPred Linear Epitope, (**B**) Chou & Fasman Beta-Turn Prediction, (**C**) Emini Surface Accessibility Prediction, (**D**) Karplus & Schulz Flexibility Prediction, (**E**) Kolaskar & Tongaonkar Antigenicity, (**F**) Parker Hydrophilicity Prediction.

**Figure 3 microorganisms-12-01270-f003:**
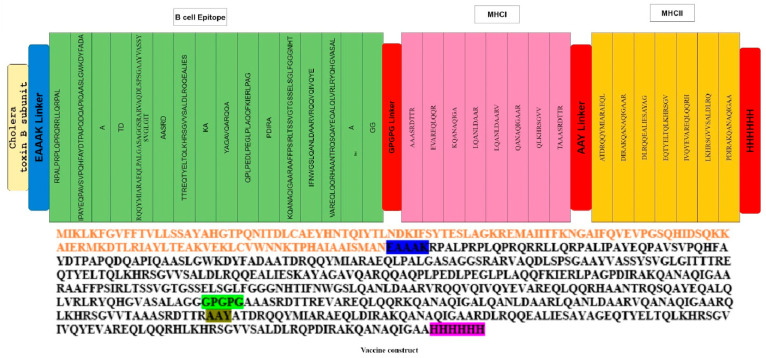
The final multi-epitope vaccine peptide is represented schematically as a 350-long amino acid sequence. The N-terminal and C-terminal ends of the peptide feature adjuvants (depicted in red), and each adjuvant is connected to the multi-epitope sequence through EAAAK linkers (highlighted in blue).

**Figure 4 microorganisms-12-01270-f004:**
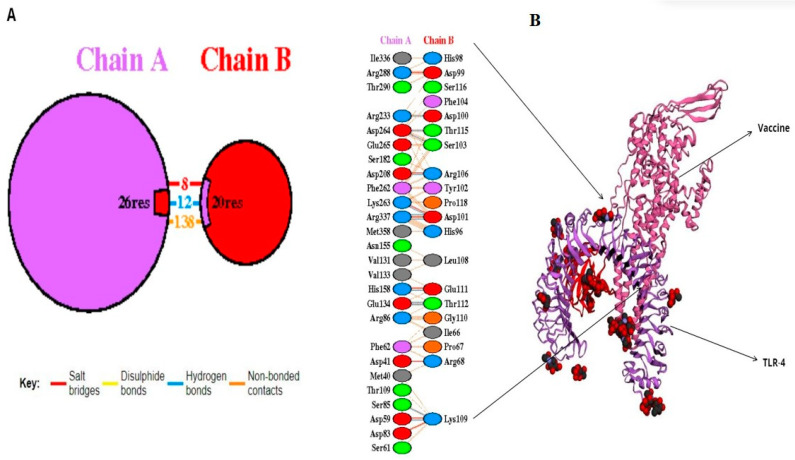
Docked vaccine construct with sbk8. (**A**) Docked vaccine (red) and sbk8 (purple). (**B**) Interaction occurs between the vaccine model and the sbk8 protein.

**Figure 5 microorganisms-12-01270-f005:**
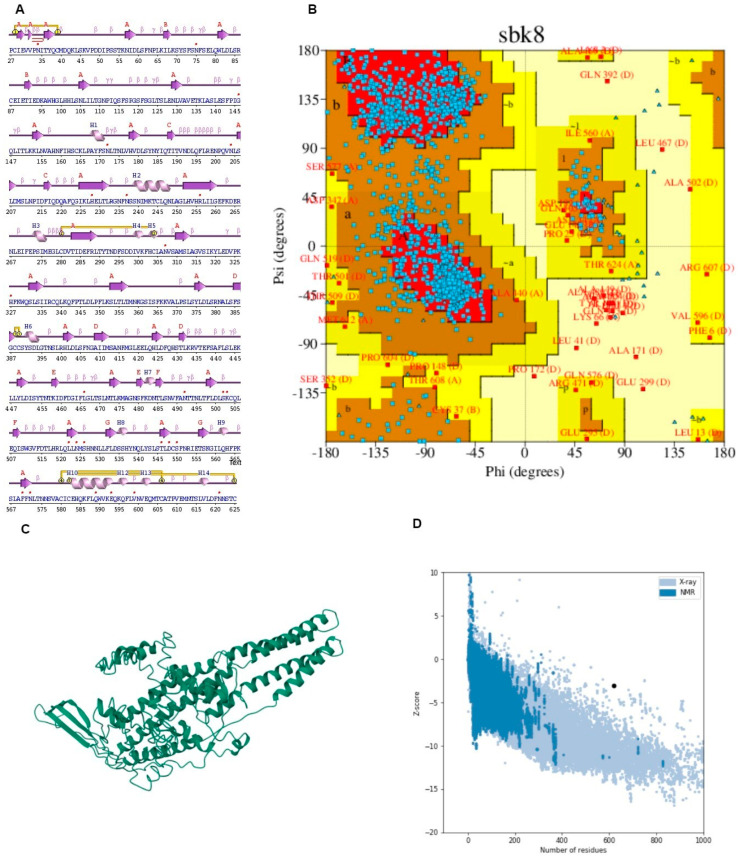
(**A**) Secondary structure predicted through PDB-sum. (**B**) Ramachandran plot depicting the validation of the structure along with the graph presenting Z-score. (**C**) I-TASSER (**D**) Graph of solubility.

**Figure 6 microorganisms-12-01270-f006:**
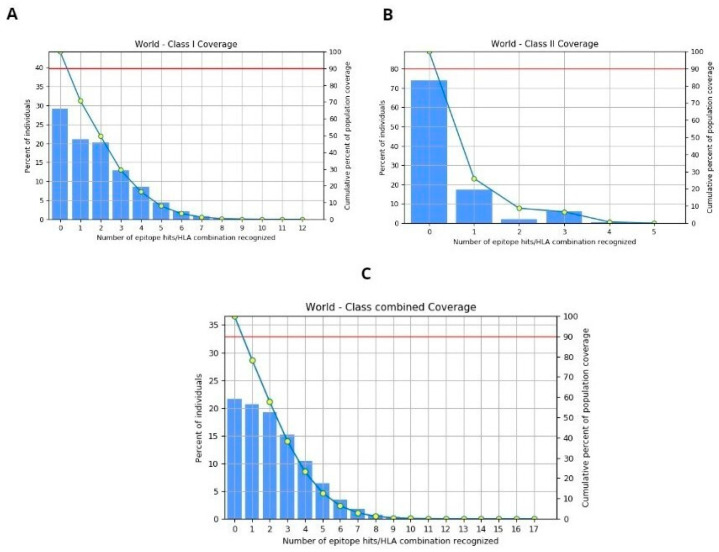
(**A**) World population coverage for MHC-I. (**B**) World population coverage for MHC-II. (**C**) Combined world population coverage for HLA allele recognized as T-cell epitope.

**Figure 7 microorganisms-12-01270-f007:**
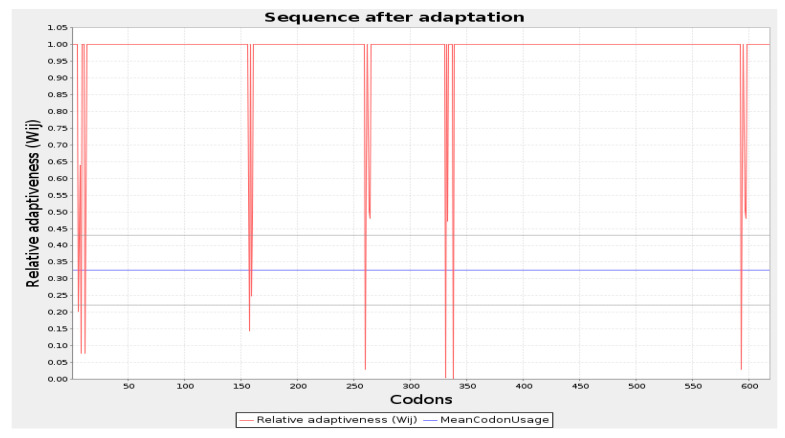
Codon optimization of the vaccine construct Final Vaccine. Here, CAI of the optimized codon and average GC content were 0.94 and 56.09%, respectively.

**Figure 8 microorganisms-12-01270-f008:**
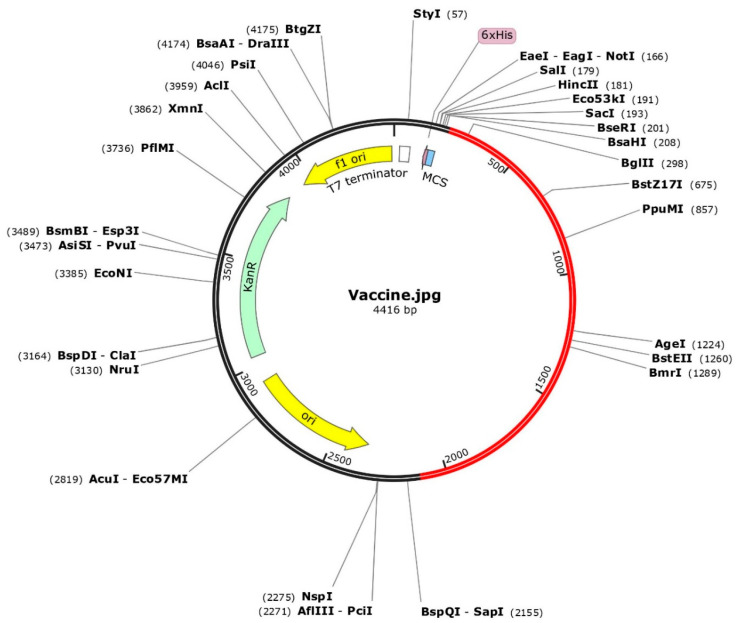
The optimized codons encoding the vaccine protein were in silico cloned into the pET28a (+) vector to facilitate expression in microbial systems. The DNA sequence was inserted into the multiple cloning site of the cloning vector. In the representation, the red portion represents the gene sequence of the designed vaccine construct, while the black portion represents the backbone of the vector. Colored arrows indicate the location and direction of gene expression, with green denoting the kanamycin resistance gene.

**Figure 9 microorganisms-12-01270-f009:**
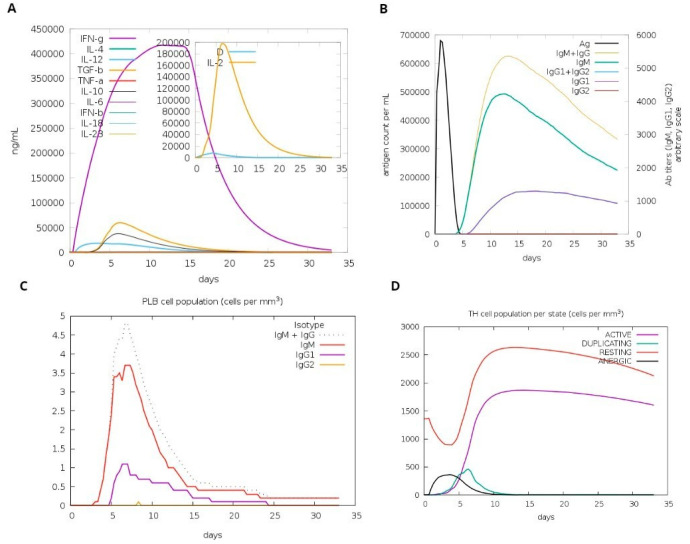
Host immune system computational modeling against designed vaccination. (**A**) Graph illustrating the synthesis of immunoglobulins. (**B**) Graph illustrating interleukins and cytokines’ induction. (**C**) The tally of plasma B lymphocytes is categorized according to their isotypes (IgM, IgG1, and IgG2). (**D**) The population of B lymphocytes by entity-state.

**Figure 10 microorganisms-12-01270-f010:**
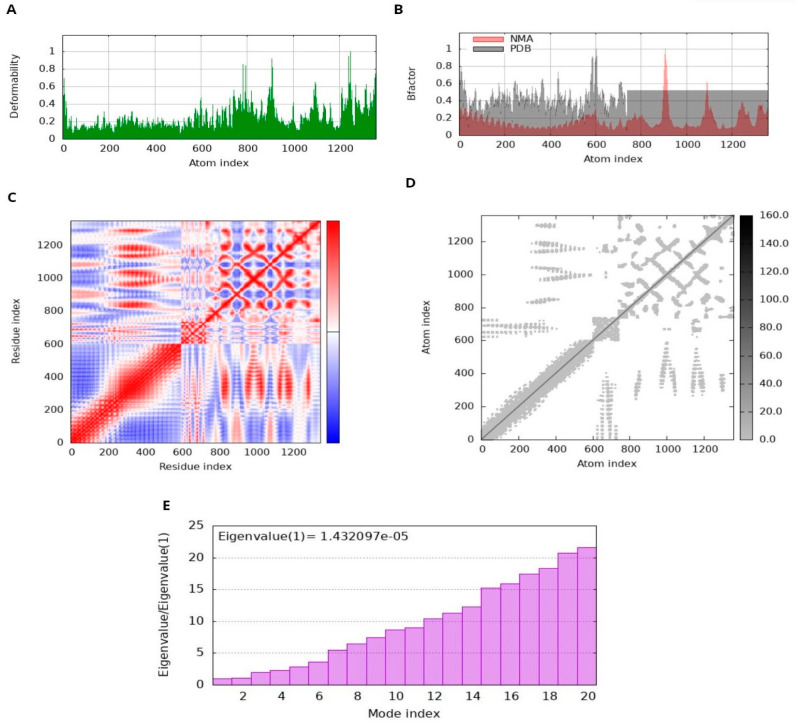
Molecular dynamics (MD) simulation of the docked MEV complex. (**A**) deformability; (**B**) B factor; (**C**) Coherence Index; (**D**) elastic network analysis; (**E**) eigenvalue.

**Table 1 microorganisms-12-01270-t001:** Five core proteins having transmembrane helices less than 1 or zero.

Accession No.	Protein Name	Sequence	Instability Index (II)
A0A2A2AF83	Glutamate dehydrogenase	MFRVCWVDDAGKVQVNRGYRIQHSMAIGPYKGGLRFHPSVNLSVLKFLAFEQTFKNALTT LPMGGGKGGSDFDPKGKSQGEIMRFCQAFAAELFRHVGADTDVPAGDIGVGGREIGYIAG YIKKLTNRADCVLTGKPLNLGGSLIRPEATGYGTVYFAEAMLNEARNSFQGMKVAVSGSG NVAQYAIEKAMALGATVVTCSDSNGTVYDPAGFSPEKLAILMDIKNVQYGRVKDYAAKVGAEYIEGKRPWHIPAEVALPCATQNELDESDAKTLIANGVVCVAEGANMPCTIEAAKAFEA AGVLFAPGKASNAGGVATSGLEMSQNAMRLAWTRQEVDQRLFGIMQSIHQACLQYGRRADGKVSYIDGANIAGFVKVADAMLAQGVV	32.64
A0A2A2AIP1	Outer membrane protein assembly factor BamA	MNHLLKRFSMRAAAAVTACACSLPLWAIEPFMVRDIRVEGLQRVEPGTVFASLPIRVGDQ YTDDKASDSIRALFALGLFNDVRIEADGNVLVVVVQERPVIGSVEFAGTREFDREALVNA MRDVGLAEGRPFDRSLADRAEQELVRQYLSRSYYGAQVVTTVTPVDRNRVNLVFTVTEG VAKIGDIRFTGNKAFSESTLKGLMDQDTGGWLSWYTKSNRYSRAKLNADLETVRSHYLQR GYLDFRIDSTQVAISPDKQSISITINVHEGERFAVSGVRLEGDYLGRDDEFKSLVQIRPG QPFNEEEVTNTREAFTEHFGNYGYAFARVQAQPEVDRERNTVAIVLRSEPQARVYVRRIN VTGNTRTRDEVIRREFRQYEASWYDGEKIRLSRDRVDRLGYFTQVEVETQEVPGVPDQVD LQITVAEKPTGSVQIGAGFSSADKLSFSFALKQENFMGSGHYLGVDLNTSKYNRTLVFST TDPYFTQSGVSRTLDAYYRTDKPYDRMGGNYSLVTYGGALRFGVPFSEIDTVYFGLGVER NRIKPGTAIPAAYLHYADTFGYSSTSVPFTIGWSRDSRDSALAPNEGRYQRFNSEWSFAG DTRYLKSNYQYQQYLPLSKRYTLAFNGELGWGKGFSGQPFPVFKNFYSGGLGSVRGFEQG TLGPRDLIGASLGGAKKVNMNVEFITPFPGAGNDRTLRVFAFVDAGNVFGEHEKVSFSDL RASTGLGLSWISPVGPLRIAYAHPIRKKAGDRIEEIQFQIGTSF	30.72
A0A2A2AHJ4	Multidrug transporter	MNPNRPALPRPLQPRQRRLLQRPALGALAALAVAAALPGCAMIPAYEQPAVSVPQHFAD TPAPQDQAPIQAASLGWKDYFADARLHRLIELALARNTDLRKAALNAEAVRQQYMIARAE QLPALGASAGGSRARVAQDLSPSGAAYVASSYSVGLGITAYEIDLFGKLRSASEAALQY LGSAASRDSAHLALVAAVAKAYFNERYAQQAMALAQSVLTTREQTYELTQLKHRSGVVSALDLRQQEALIESAKADYAGAVQARQQALNALAMLINQPLPEDLPEGLPLAQQFKIERLPA GLSSEVLLNRPDIRAAEFALKQANAQIGAARAAFFPSIRLTSSVGTGSSELSGLFGGGNH TWSFAPAITLPIFNWGSLQANLDAARVRQQVQIVQYEGAVQAAFQDVANALVAREQLQQR HAANTRQSQAYEQALQLVRLRYQHGVASALDLLDAERSSYAANMALLANQLTQLENLADLYKALGGGLKP	49.29
A0A2A2AJA1	TonB-dependent siderophore receptor	MPPNANHSLLRESAQAVGGLDSSRKGPHTSGGPGQRPLPRPMPGSMPAGATHGAAVRAGL LQHIAQHGGIAMTAQLVSRPWQRRVFSSLSSAPARPPRWVPGALAAAVLWALAAWPAGAAPAKAAPDAAQPAPAGPGRQPVAELPTVTVRGDAPDAPGGAAADAGRSGKLARRALGATKTDTPLLHVPQSVSVITETALRDSGATSLDQALAYHPGLYAPVGGGNDSSRYDFVSLRGQSYNGAMFFDGMRASFGVGNLSLPQFDPWLIERVEVLRGPASALYGQGLPGGLVNLRAKRPGTQAHRAAGLTLGSHGQRALRLDAGGQAGQGALDWRLAALARAAGNRIAHVREQRVALAPSLRWRIAPGTALTLLASHQRDPKGGYYHSALPLQGTLTPLPGGGHIPRRFFVGEPGFDRFARRQSTAGYDFEHALGQGWQLRHTLRAIDSQAEVQALSATALVPPATLMRSAMAVHSRTRALLSDTTVQGRVQTGAAQHRLLLGLDAMRSRTHQRLGMNLQGLPPIDIWQPGYGQAIAVPEGPGSAMLWADTRDRASQIGLYAQDQIDWGRWRLTLGGRYDIARSRSAREGRLMGVLPTDAASRQRDRAFSGRAALGYQLGDALAAYLSHGSAFLPQTGLDARGQGYRPLTARQWEAGLKYAPPQGGLQLAAAIFQIQQKNALTPDPEPSHVCPGLAGPGACMVQTGRQRTRGLELEAQAELGRASFVHASLTLLDARITASNGPEQGQRPVNIPARTASFWLDHALSPQWRMSLGLRHTGSTRADPANTVHVPGHTLMDAALRYRFGHGGSHDGASAERPSLTLRASNLADRRYVSCASAS YCNWGRGRTLSLELHYPW	47.30
A0A2A2APV4	Type IV pilus biogenesis/stability protein PilW	MKPRHTLWRLPAALALAAAALGLTACQTSYTRSSVPVATPGAAAAASEPDVQRAAKVRLE LASEYLRVGRSNVALEEINHVLSIAPHMVEAYMLRGMIHADQHNFAAAEADYARVMRERGNDPDALHNYGWILCRQGRYADAEGYFDRVLAAPGYTASARTLMAKGLCQQSAGKAGAAMATLQRAYEVDPNNPIVAYNLASMLYHAGRVADAQTYLRRLNGSDLANAETLWLGIKVENALGRRDRVRELGSVLAHRFPNSREFALYERGAFYE	34.87

**Table 2 microorganisms-12-01270-t002:** Epitope prioritization to select epitopes with the potential to be incorporated into the vaccine construct by applying various filters Top of Form.

Proteins	B-Cell Epitopes	MHC-II	MHC I	Allergenicity	Antigenicity	Toxicity	Solubility
>tr|A0A2A2AHJ4|A0A2A2AHJ4_9BURK Multidrug transporter OS = Vandammella animalimorsus	RPALPRPLQPRQRRLLQRPAL	ATDRQQYMIARAEQL	AAASRDTTR	Nonallergen	ANTIGEN	NON-TOXIN	Soluble
	IPAYEQPAVSVPQHFAYDTPAPQDQAPIQAASLGWKDYFADA	DIRAKQANAQIGAAR	EVAREQLQQR	Nonallergen	ANTIGEN	NON-TOXIN	Soluble
	A	DLRQQEALIESAYAG	KQANAQIGA	Nonallergen	ANTIGEN	NON-TOXIN	Soluble
	TD	EQTYELTQLKHRSGV	LQANLDAAR	Nonallergen	ANTIGEN	NON-TOXIN	Soluble
	RQQYMIARAEQLPALGASAGGSRARVAQDLSPSGAAYVASSYSVGLGIT	IVQYEVAREQLQQRH	LQANLDAARV	Nonallergen	ANTIGEN	NON-TOXIN	Soluble
	AASRD	LKHRSGVVSALDLRQ	QANAQIGAAR	Nonallergen	ANTIGEN	NON-TOXIN	Soluble
	TTREQTYELTQLKHRSGVVSALDLRQQEALIES	PDIRAKQANAQIGAA	QLKHRSGVV	Nonallergen	ANTIGEN	NON-TOXIN	Soluble
	KA		TAAASRDTTR	Nonallergen	ANTIGEN	NON-TOXIN	Soluble
	YAGAVQARQQA			Nonallergen	ANTIGEN	NON-TOXIN	Soluble
	QPLPEDLPEGLPLAQQFKIERLPAG			Nonallergen	ANTIGEN	NON-TOXIN	Soluble
	PDIRA			Nonallergen	ANTIGEN	NON-TOXIN	Soluble
	KQANAQIGAARAAFFPSIRLTSSVGTGSSELSGLFGGGNHT			Nonallergen	ANTIGEN	NON-TOXIN	Soluble
	IFNWGSLQANLDAARVRQQVQIVQYE			Nonallergen	ANTIGEN	NON-TOXIN	Soluble
	VAREQLQQRHAANTRQSQAYEQALQLVRLRYQHGVASAL			Nonallergen	ANTIGEN	NON-TOXIN	Soluble
	A			Nonallergen	ANTIGEN	NON-TOXIN	Soluble
	GG			Nonallergen	ANTIGEN	NON-TOXIN	Soluble

**Table 3 microorganisms-12-01270-t003:** Top models of docked complexes of designed vaccine with TLR-4.

Cluster	Members	Representative	Weighted Score
0	82	Center	−1086.4
		Lowest Energy	−1221.2
1	51	Center	−802.5
		Lowest Energy	−972.1
2	51	Center	−1065.5
		Lowest Energy	−1065.5
3	48	Center	−874.5
		Lowest Energy	−912.2
4	30	Center	−802.4
		Lowest Energy	−962.9
5	29	Center	−1078.4
		Lowest Energy	−1078.4
6	26	Center	−826.3
		Lowest Energy	−945.9
7	22	Center	−817.1
		Lowest Energy	−916.9
8	19	Center	−798.2
		Lowest Energy	−862.2

**Table 4 microorganisms-12-01270-t004:** Refined models of the designed vaccine.

Model	GDT-HA	RMSD	MolProbity	Clash Score	Poor Rotamers	Rama Favored
Initial	1.0000	0.000	3.744	25.7	19.7	67.5
MODEL 1	0.8811	0.619	2.616	26.6	1.3	86.7
MODEL 2	0.8762	0.642	2.551	26.9	0.8	86.4
MODEL 3	0.8782	0.634	2.574	26.9	1.1	86.0
MODEL 4	0.8762	0.630	2.644	28.0	1.3	86.4
MODEL 5	0.8786	0.631	2.585	30.0	0.6	86.9

## Data Availability

The raw data supporting the conclusions of this article will be made available by the authors on request.

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
