# Peer review of "Immunoinformatics and Reverse Vaccinology Approach for the Identification of Potential Vaccine Candidates against Vandammella animalimors"

_microorganisms, 2024, doi:10.3390/microorganisms12071270_

Round 1

Reviewer 1 Report

Comments and Suggestions for Authors

The objective of the study was the exploration of the proteome of Vandamella animalimorsus.

The authors must add in the title that the study was performed in silico, as readers may be misled to think that an actual proteomics technology was employed in the work.

The authors must state clearly what gaps in the literature would be filled by the results of the study.

What will this publication add in the international literature in comparison to other similar publications?

Also, please present in a separate sub-section of M&M, the precise control procedures that were employed. For example, did you compare with the proteomes of other bacteria? Perhaps, did you use various approaches to gather information about the proteomes of the organism? Did you take into account in your study the type strain of the organism?

Visualization. The tables and the figures of the manuscript are appropriate for the manuscript.

References. Please note that there is a recent (March 2024) relevant reference and please care to cite it in the revised manuscript and use it in the Discussion.

Conclusions. The concluding section is not fully consistent with the findings, so please rewrite and please tone down to make it of appropriate standard.

Patent. Do you intend to obtain a commercial patent for this results? If yes, please include the reference number of the patent in the manuscript.

Overall. Please revise major taking into account all the above comments.

Author Response

Response to Reviewer 1

The authors must add in the title that the study was performed in silico, as readers may be misled to think that an actual proteomics technology was employed in the work.

Respected reviewer, Thank you very much for your kind suggestions. We would like to explain with humble that the terminologies Immunoinformatics and reverse vaccinology in the title explain quotes the computational work.

The authors must state clearly what gaps in the literature would be filled by the results of the study.

Thank you for your constructive comments. We have revised the suggestions and we anticipate that it will be fine.

What will this publication add to the international literature in comparison to other similar publications?

Thank you for your constructive comments. We have revised and incorporated suggested information in the Conclusions. 

Also, please present in a separate sub-section of M&M, the precise control procedures that were employed. For example, did you compare with the proteomes of other bacteria? Perhaps, did you used various approaches to gather information about the proteomes of the organism. Did you take into account in your study the type strain of the organism?

Respected reviewer, Thank you very much for your very constructive queries. We have revised materials and methods and we anticipate that it will be fine now.

Visualization. The tables and the figures of the manuscript are appropriate for the manuscript.

Thank you for your compliments

References. Please note that there is a recent (March 2024) relevant reference and please care to cite it in the revised manuscript and use it in the Discussion.

Respected reviewer, Thank you very much for your very constructive queries. Although we could not get into a specific article in March 2024 i.e. Epitope-based vaccine or about bacterium.  We have revised the references and have added some references such as Muggui et al., 2024 related to bacterium updates that it will be fine now.

Conclusions. The concluding section is not fully consistent with the findings, so please rewrite and please tone it down to make it of an appropriate standard.

We have revised it and we anticipate that it will be fine now.

Patent. Do you intend to obtain a commercial patent for these results? If yes, please include the reference number of the patent in the manuscript.

Thank you very much for your kind guidance. At the moment it is not in process,

Overall. Please revise the major taking into account all the above comments.

Thank you very much for your kind guidance. We have tried our best to address all the major comments and we anticipate that it will be fine now.   

Reviewer 2 Report

Comments and Suggestions for Authors

The authors reported an in silico approach for identification of a vaccine candidate against bacterium V. animalimors. After careful reading, this reviewer found many weaknesses of the present study. Abstract should have one sentence on the significance of this bacterium, to justify the need of such a study. The entire Introduction is redundant and lacks coherence. It requires a complete re-writing. There is no connection between paragraphs in this section. Justification of the study is missing. No information of the specific background and if any such study was previously undertaken. 

The Methods section severely lacks the necessary information and is written vaguely. Methodology require a comprehensive re-writing with detailed information on the protocols and methods used, with justification. 

Result and Discussion section also lacks coherence and needs to be presented with better graphics. Figure 3 is a poor quality image. Other figures also lack required dpi (are not 300 dpi) and are poor quality. The authors failed to justify the study and its novelty. 

Other specific comments: 

L50: There is no connection between this paragraph and the previous paragraph. Please provide adequate background information before switching from virus-borne rabies infection to the bacterial transmission. 

Figure 1: Font size could be increased for better reading. Also, citations required if previously reported methods were followed. 

L95-108: Methods: Enough details are not provided. How was BLASTp performed. Statistics for confidence?

L109-121: Methods: Detailed methodology should be provided. The current version skims through the protocols used. 

L123: "The protein sequences that were shortlisted and supposed final, meeting the criteria for potential candidates in multi-epitopes peptide V. animalimorsus vaccine,". This is vaguely written and provides no required information on the criteria. 

L143-146: Methods: There is no information on why these linkers were chose over others. 

L185-188: "In our investigation, we used the iMODS server for this purpose [34]. We used the iMODS server for this reason in our experiment [35]. The iMODS (NMA) server provides insightful information on the fundamental dy -namics of many proteins [36]." This provides no information and also is poorly written. 

L188: "Depicting protein dynamics and stability using several metrics, including eigenvalues, covariance, B-factors, and deformability [37]." Incomplete sentence. 

The overall quality of the study and manuscript is poor. 

Comments on the Quality of English Language

English can be improved at many places. 

Author Response

Response to Reviewers 2

The authors reported an in-silico approach for identification of a vaccine candidate against bacterium V. animalimors. After careful reading, this reviewer found many weaknesses of the present study.

Respected reviewer, we greatly obliged your kindness for constructive comments. We have revised Introduction and we anticipate that it will be as per your kind suggestions. 

Abstract should have one sentence on the significance of this bacterium, to justify the need of such a study.

We have revised as per your kind suggestions and we anticipate that it will be as per your kind suggestions. 

The entire Introduction is redundant and lacks coherence. It requires a complete re-writing. There is no connection between paragraphs in this section. Justification of the study is missing. No information of the specific background and if any such study was previously undertaken. 

We have revised entire introduction and the information incorporated for the synchronizations of each paragraph can be found in track changes.

The Methods section severely lacks the necessary information and is written vaguely. Methodology require a comprehensive re-writing with detailed information on the protocols and methods used, with justification. 

We are really thankful for your guidance.  We have revised entire method section and we anticipate that it will be as per your kind suggestions.

Result and Discussion section also lacks coherence and needs to be presented with better graphics. Figure 3 is a poor-quality image. Other figures also lack required dpi (are not 300 dpi) and are poor quality. The authors failed to justify the study and its novelty. 

We are really thankful for your constructive comments.  We have revised entire section of Results and Discussion and also have improved Images quality.  We anticipate that it will be as per your guidance.

Other specific comments: 

L50: There is no connection between this paragraph and the previous paragraph. Please provide adequate background information before switching from virus-borne rabies infection to the bacterial transmission. 

We have revised it

Figure 1: Font size could be increased for better reading. Also, citations required if previously reported methods were followed. 

Thank you for constructive comments. We have revised it

L95-108: Methods: Enough details are not provided. How was BLASTp performed. Statistics for confidence?

Thank you for constructive comments. We have revised it.

L109-121: Methods: Detailed methodology should be provided. The current version skims through the protocols used. 

Thank you for constructive comments. We have revised it.

L123: "The protein sequences that were shortlisted and supposed final, meeting the criteria for potential candidates in multi-epitopes peptide V. animalimorsus vaccine,". This is vaguely written and provides no required information on the criteria. 

Thank you for constructive comments. We have rephrased it with relevant information.

L143-146: Methods: There is no information on why these linkers were chose over others. 

Respected reviewers, although a variety of linkers are available and used in various studies but the linkers used in this study have better flexibility, and structural stability with reduced immunogenicity and protease resistance compared to other linkers.

L185-188: "In our investigation, we used the iMODS server for this purpose [34]. We used the iMODS server for this reason in our experiment [35]. The iMODS (NMA) server provides insightful information on the fundamental dy -namics of many proteins [36]." This provides no information and also is poorly written. 

Thank you for your constructive comments. We have revised it and we anticipate that it will be as per your kind suggestions.

L188: "Depicting protein dynamics and stability using several metrics, including eigenvalues, covariance, B-factors, and deformability [37]." Incomplete sentence. 

Thank you for your constructive comments. We have revised it and we anticipate that it will be as per your kind suggestions.

The overall quality of the study and manuscript is poor. 

Respected Reviewers, we really really appreciate and acknowledge your very useful guidance which helped us to improve the manuscript extensively. We strongly believe that we have followed your suggestions to meet the standard of publications.

Reviewer 3 Report

Comments and Suggestions for Authors

The authors have carried out an excellent piece of research, very important and relevant to the area of interest.

Reverse vaccinology has grown as an excellent way of identifying vaccines against vectors and pathogens. 

Minor revisions are needed: 

Figure1:  MHC instead of MCH in this figure and all over the text.

Material and Methods 2.7: have an issue in the format

Table 1: describe better in the legends what is the table.

Figure 3: The image is a little blurry and it's not possible to understand the sequence below.

Figure 5: Have a problem with format also. 

Author Response

Response to Reviewers 2

The authors have carried out an excellent piece of research, very important and relevant to the area of interest. Reverse vaccinology has grown as an excellent way of identifying vaccines against vectors and pathogens.

Respected reviewer, Thank you very much for your kind consideration and compliments. We have revised the manuscript entirely and we anticipate that it has been improved.

Minor revisions are needed: 

Figure 1:  MHC instead of MCH in this figure and all over the text.

Thank you very much for constructive comments. We have revised Figure as well as entire manuscript.

Material and Methods 2.7: have an issue in the format

We have revised the suggested part of Materials and Methods

Table 1: describe better in the legends what is the table.

We have revised it

Figure 3: The image is a little blurry and it's not possible to understand the sequence below.

 We have revised it

Figure 5: Have a problem with format also.

We have revised it 

Round 2

Reviewer 1 Report

Comments and Suggestions for Authors

All issues resolved; no further comments.

Reviewer 2 Report

Comments and Suggestions for Authors

No further comments.